# Factors affecting the purchase behaviour of farmers' markets consumers

**Hakan Adanacioglu** [ORCID] *

Department of Agricultural Economics, Faculty of Agriculture, Ege University, Izmir, Turkey

* hakan.adanacioglu@ege.edu.tr

**Data Availability Statement:** All relevant data are within the manuscript and its Supporting Information files.

**Funding:** This research was funded by the Ege University Scientific Research Projects Coordination Unit, grant number 15-ZRF-049.

## Abstract

The main purpose of this study is to determine the factors that motivate consumers who shop at farmers' markets. The data for this study were gathered from questionnaires of 363 consumers from eight farmers' markets in seven districts of Izmir province, Turkey. To reveal the consumer profile of the farmers' markets examined in this study, consumer segments were determined using factor and cluster analysis. Two different consumer segments—'conventional' and 'conscious'—were identified in the farmers' markets examined. 'Conventional Consumers' reflect typical consumer behaviours and give more importance to factors such as the location of and access to the market, quality and freshness of the products, activities at and around the market and the availability and variety of products. 'Conscious Consumers', in contrast, represent a group that is more sensitive about food safety. The majority of consumers (63.64%) who visited farmers' markets were from the Conscious Consumer segment. The majority of the consumers who visit farmers' markets are conscious consumers, requiring the strategies related to these markets to be revised. Farmers' markets should be improved in terms of selecting vendors, food safety, physical facilities and social activities.

## Introduction

Farmers' markets—indoor or outdoor marketplaces where producers sell the products they produce directly to consumers—are one of the most common options for direct marketing. A farmers' market is defined as a common area wherein producers sell fresh fruits and vegetables as well as other agricultural products directly to consumers at separate stands [1]. Farmers' markets are one of the activities in alternative food networks. In many developed countries, farmers' markets develop for numerous reasons, such as enabling an increase in farmer income, benefiting consumers by providing fresh, high-quality products and facilitating communication and social interaction with local communities [2]. Farmers' markets are viewed as an opportunity to support small farmers, promote direct communication between consumers and producers, ease access to local, healthy products, support local economies and offer consumers an alternative to industrial food products [3]. Important factors such as an increase in consumer demand for fresh and locally grown products, changes in the agricultural economy,

https://bap.ege.edu.tr/eng-/Homepage.html The funders had no role in study design, data collection and analysis, decision to publish, or preparation of the manuscript.

**Competing interests:** The authors have declared that no competing interests exist.

consumers' interest in direct interaction with producers and efforts to access product resource information are considered reasons for the increase in the number of farmers' markets [4].

When previous studies are analysed, it can be seen that the two most common factors that motivate consumers to shop at farmers' markets are product quality and freshness. In several studies, quality has been identified as one of the most important attributes motivating consumers to shop at farmers' markets [4–13]. González-Azcárate et al. [14] also stated that food quality was the main consumer driver to get involved in short food supply chains. Additionally, many studies have indicated that product freshness is one of the main attributes that motivates consumers to shop at farmers' markets [6–8, 10, 15–23].

According to the aforementioned studies, health and food safety concerns are also major factors that motivate consumers to visit farmers' markets [12, 17–19, 24]. Some consumers believe that no chemicals are used in products sold at farmers' markets. These consumers visit these markets to obtain organically produced and non-genetically modified organism (GMO) products. According to the findings of previous studies, other important factors that motivate consumers to shop at these markets are accessing locally produced foods [4, 8, 15–18] or foods of domestic origin [25], supporting local farmers by purchasing products directly from producers [5, 7, 10, 12, 16, 18, 26], having more options in terms of variety [7, 15, 21, 22], gaining access to special products [9, 10], communicating directly with farmers [10, 11, 26], getting ready-made meals [23, 27] and enjoying a fun, friendly atmosphere [5, 19, 20, 23, 27]. In previous research findings, it has been revealed that the primary motivating factors that encourage consumers to visit these markets are based on non-economic reasons.

In some previous studies, the factors affecting the decisions of consumers to shop at farmers' markets were examined by consumer segments using cluster analysis [10, 28, 29]. In a study conducted by Hunt [10] in the US (i.e. Maine), consumers who shopped at farmers' markets were grouped into three segments—'Lifestylers', 'Seasonal Shoppers' and 'Utilitarians'. In another study conducted in the US (i.e. Indiana and Illinois), Arrington et al. [28] identified four consumer segments of farmers' markets—'Recreational' (42%), 'Minimalists' (27%), 'Enthusiasts' (23%) and 'Time challenged' (8%). In another study conducted by Elepu and Mazzocco [29] in the US (i.e. Chicago and Metro East-St. Louis), five consumer segments were defined—'Market Enthusiasts', 'Recreational Shoppers', 'Serious Shoppers', 'Low-involved Shoppers' and 'Basic Shoppers'.

Previous studies on farmers' markets mostly focused on North America. Additional research is required that examines farmers' markets in different parts of the world. It is especially important to examine countries wherein farmer markets are just beginning to develop. Turkey is one of the countries in which farmers' markets have begun to become widespread in recent years. Statistical data on farmers' markets are not officially published by the Turkish government; however, it is possible to compile some information by considering the farmers' markets examined within the scope of this study. Farmers' markets in Turkey are usually established by local municipalities in districts around the province and usually comprise small farmers. Only fruit and vegetables are sold in small-type farmers' markets. In large-type farmers' markets, many stands sell ready-made foods, noodles, traditional soups, tomato paste, honey, handicrafts and souvenirs as well as fruit and vegetables.

The main purpose of this study is to determine the factors motivating consumers who shop at farmers' markets. The factors that motivate consumers to shop at farmers' markets were examined according to consumer segment. Previous studies indicate that there are various types of customer segments shopping at farmers' markets [10, 28, 29]. Considering previous studies, the following research hypothesis was proposed:

**H1.** *The factors that motivate consumers to shop at farmers' markets vary according to various consumer segments.*

A face-to-face survey was conducted with customers at eight farmers' markets in the Izmir province of Turkey to determine the factors motivating consumers to shop at farmers' markets. Knowing the motivating factors, consumer segments, consumer preferences and expectations that attract consumers to farmers' markets can significantly benefit the producers in these markets as well as those who manage them. Thus, those who manage farmers' markets can ensure that consumer demands are met more effectively. This might lead to an increase in the number of consumers patronising existing farmers' markets and an increase in the demand for new farmers' markets. In this way, local administrators will have an opportunity to provide economic benefits to their region, support local agricultural producers and offer healthy and high-quality food to consumers. Simultaneously, local administrators can establish more fluent relationships with consumers by learning their opinions about the producers and market-related subjects (i.e. products, activities, etc.). Additionally, local managers having access to consumer profiles of those who patronise farmers' markets will enable them to better plan the locations of possible farmers' markets in the future.

## Materials and methods

### Study site and data collection

The primary data source for this study was surveys of consumers who patronise farmers' markets in seven districts of Izmir, Turkey. In selecting the farmers' markets where the research was conducted, the active working status of the markets and the fact that the vendors were producers were considered. In this context, eight farmers' markets in seven districts, including Seferihisar, Urla, Çeşme, Foça, Aliağa, Bergama and Bornova, were included in the research area. Two of the eight farmers' markets included in the research are located in the Seferihisar district (i.e. Merkez and Sığacık).

In determining the number of consumers (i.e. the sample size) to interview in the identified farmers' markets, the 'Non-clustered One Stage Simple Random Probability Sampling Based on Main Audience Ratios' method was used [30]. The fact that the number of consumers shopping at farmers' markets, or the main audience, was not fully known made this method effective. Nakip [31] stated that the estimation of the sample volume in marketing research can be done using ratios. According to Nakip [31], the biggest advantage of determining the sample volume using ratios is calculating the sample volume without the need for standard deviations. The sample volume formula (1) according to this method is as follows:

$$n = \frac{t^2 \times p \times q}{e^2} \tag{1}$$

where
n: sample volume
t: table value corresponding to 95% significance (1.96)
p: probability of the examined event occurring within the main audience (i.e. the possibility of consumers shopping at farmers' markets; taken as 50% to reach the maximum sample size)
q: the probability that the event under consideration will not to occur (i.e. the possibility of consumers not shopping at farmers' markets; 1—p)
e: accepted margin of error (margin of error was taken as 5%)

According to the aforementioned above, the sampling volume was calculated as 384. While the calculated sample volume was distributed according to the farmers' markets in the identified districts, the number of producers, size of the market area and shopping volume (based on observation) were taken as the base. Within this framework, the plan was to interview 30 people or less at small-scale farmers' markets, 31–49 people at medium-scale markets and at least

50 people or more at large-scale farmers' markets. Considering the sample volume and the criteria determined for the field study, 384 surveys of consumers were conducted. However, 21 of these surveys wherein the researchers were given incomplete information and data reliability was suspected were not included in the data analysis. In this context, a total of 363 surveys were considered for data analysis of eight farmers' markets. Accordingly, the distribution of 363 consumer surveys at local farmers' markets in the districts is as follows: Aliağa (15), Bergama (31), Bornova (65), Çeşme (53), Foça (41), Seferihisar-Merkez (64), Seferihisar-Sığacık (45) and Urla (49). The survey was conducted in September and October of 2016 and interviews with consumers were performed at farmers' markets. Great care was taken to ensure that the consumers participating in the survey were individuals who were interested in farmers' markets and had previously visited these markets in order to increase the reliability of the answers.

A structured questionnaire form was used to collect data on purchasing behaviors of farmers' markets consumers. The questionnaire was designed to collect data covering in-formation on demographic and behavioral characteristics of farmers' market shoppers, products sold at the farmers' markets, motivating factors for shopping at farmers' markets, and perceptions of shoppers towards farmers' markets.

## Ethical statement

This study, whiFch was carried out between 2015–2019, was reviewed and approved by Ege University Scientific Research Projects Coordination Unit. However, this study does not include an ethics committee approval before doing it. In 2015, when the project was initiated, the Scientific Research and Publication Ethics Committee of Ege University did not legally oblige to obtain an ethics committee report for survey research involving human participants. Obligation to obtain ethical committee approval for research involving data acquisition from human participants started in 2017. On the other hand, this survey research was carried out in 2016.

The survey participants were invited to take part in a research study about purchasing behaviors of farmers' markets consumers. They were first fully informed as to the intent and purpose of the study. They were asked for their consent to participate in this study. The consent procedure been carried out face-to-face.

## Statistical analysis

The responses of the consumers to the five-point Likert-type scale questions were tested with reliability analysis using Cronbach's alpha coefficient. The Cronbach's alpha coefficient is obtained by dividing the sum of the variances of the questions in one scale by the general variance. This coefficient takes a value between zero and one. If this coefficient is higher than or equal to 0.6, the scale is accepted as reliable [32].

The chi-squared test of independence was used to determine whether the consumer segments differed in terms of their demographic characteristics. The chi-squared test of independence is used to analyse whether there are relationships between two or more groups of variables. For the chi-squared test of independence to be applied, the results of the observations must be presented as classified or grouped compound data series [33].

## The methods used to determine consumer segments of farmers' markets

The main hypothesis of this study is that the factors that motivate consumers to shop at farmers' markets vary according to various consumer segments. Consumer segments were

determined according to the market attributes that might affect consumers' decision to shop at farmers' markets. Data were analyzed using IBM SPSS statistics 22 software.

Factor analysis was used to identify the variables that were less meaningful and to ensure that the variables were independent from each other as there were many variables related to market attributes. The variables used in the factor analysis represent market at-tributes. Twenty-two market variables were determined for farmers' markets (see Table 2 for a list of variables). During the survey, consumers who patronised farmers' markets were asked the level of importance that they attached to each variable. For this purpose, a five-point Likert scale was used. The suitability of the data for factor analysis was deter-mined by conducting the Kaiser-Meyer-Olkin (KMO) test, which measures sample adequacy by comparing the magnitude of the observed correlation coefficients with the partial correlation coefficients. The KMO value found is evaluated as follows: below 0.50 is unacceptable, 0.50 is weak, 0.60 is moderate, 0.70 is good, 0.80 is very good and 0.90 is excellent [32].

To determine the number of factors, eigenvalue statistics were used. An eigenvalue shows the variance explained by a factor [32]. Factors with eigenvalue statistics higher than 1 are considered significant. In the rotation stage of factor analysis, the factor under which a variable has the greatest absolute value is considered to be closely related to that factor. Factor loading values or weights of 0.50 and greater are considered to be quite good [33]. Variables with a factor loading value of 0.50 or greater during the analysis of the data were included in the analysis.

In the second stage of the study, cluster analysis was performed by applying the k-means clustering method to determine the consumer segments of the farmers' market. The standardised factor scores obtained from factor analysis were used to perform k-means clustering. The k-means clustering method is one of the most commonly used non-phased clustering methods; it aims to separate clusters of continuous p-variable data from multiple (N) units into k clusters to minimise the sum of intra-cluster squares. It is not necessary to calculate the distance matrix or the similarity matrix for clustering units using the k-means method. Experimentally, it is sufficient to predetermine the possible number of clusters of data [34].

Discriminant analysis was applied to determine the number of clusters in the k-means clustering method. Discriminant analysis was used to predict group membership and classify each observation into groups based on the standardised factor scores obtained from factor analysis. The number of clusters in k = 2, 3, 4. . . was increased one by one, and Wilk's lambda values were found using discriminant analysis. Then, the number of clusters with the highest significance was accepted as the most appropriate frag-mentation [34]. The Wilk's lambda statistics show the part (rate) of the total variance in the separation scores that was not explained by the difference between the groups [33].

## Results and discussion

### Demographic characteristics of the interviewed consumers

Information on the demographic characteristics of the interviewed consumers is provided in Table 1. When the consumers who patronise farmers' markets were analysed ac-cording to gender distribution, a woman-dominated profile was encountered—65.56% of the 363 consumers interviewed were female and 34.44% were male. Many studies demonstrate that the consumers who shop at farmers' markets are generally female [5, 6, 9, 12, 20, 22, 35–37]. When the distribution of consumers shopping at farmers' markets by age group was analysed, 32.23% of consumers were in the range of 55–64 years of age and 18.18% were in the age range of 65 and over. In general, half of the consumers visiting farmers' markets were 55 years of age

**Table 1. Demographic characteristics of consumers interviewed at farmers' markets.**

| Characteristic | Category | Frequency | Percentage (%) |
|---|---|---|---|
| **Gender** | Female | 238 | 65.56 |
| | Male | 125 | 34.44 |
| **Age** | 18–24 | 7 | 1.93 |
| | 25–34 | 21 | 5.79 |
| | 35–44 | 57 | 15.70 |
| | 45–54 | 95 | 26.17 |
| | 55–64 | 117 | 32.23 |
| | 65 and over | 66 | 18.18 |
| | Average age | 52.80 (years) | |
| **Marital status** | Married | 294 | 80.99 |
| | Single | 69 | 19.01 |
| **Education** | Illiterate | 5 | 1.38 |
| | Primary school | 71 | 19.56 |
| | Secondary school | 42 | 11.57 |
| | High school | 91 | 25.07 |
| | Undergraduate | 125 | 34.43 |
| | Graduate | 29 | 7.99 |
| **Monthly household income (TRY)[a]** | 2,000 or less | 92 | 25.34 |
| | 2,001–5,000 | 179 | 49.31 |
| | 5,001–7,500 | 58 | 15.98 |
| | 7,501–10,000 | 18 | 4.96 |
| | Over 10,000 | 16 | 4.41 |
| **Household size (people)** | 1 | 45 | 12.40 |
| | 2 | 111 | 30.58 |
| | 3 | 92 | 25.34 |
| | 4 | 75 | 20.66 |
| | 5 and more | 40 | 11.02 |
| | Average household size | 2.92 (people) | |
| **Employment status** | Retired | 127 | 34.99 |
| | Housewife | 76 | 20.94 |
| | Employed | 148 | 40.77 |
| | Student | 5 | 1.37 |
| | Unemployed | 7 | 1.93 |

[a]TRY: Turkish Lira; the yearly average exchange rate of US dollar to Turkish lira for 2016 was US$1 = TRY3,02 [39].

or older. The average age of consumers was calculated as 52.80 years. Some studies have shown that the average age of farmers' market consumers is over 50 [4, 6, 36, 38].

When the education level of farmers' market consumers was analysed, almost half, or 42.22%, were found to have an undergraduate or graduate level of education. Therefore, it is possible to state that the consumers who patronise farmers' markets are generally educated. Some studies have indicated that consumers who patronise farmers' markets are generally highly educated [4–6, 9, 12, 22].

A total of 49.31% of the respondents had a net monthly household income between 2,001 and 5,000 Turkish lira (TRY) (US$662.58–1,655.63). Conversely, the ratio of those whose net monthly household income was between 5,001 and 7,500 TRY (US$1,655.96–2,483.44) was 15.98%, and the ratio of those whose incomes were over 7,500 TRY was 9.37%. The median

monthly disposable household income for Turkey in 2016 was 2,611 TRY (US$865), while the mean monthly income was 3,450 TRY (US$1,142) [40]. After a general evaluation of this data, we can determine that farmers' markets draw a middle-income group profile. The average annual household net income was between US$7,951 and 19,868, considering the segment comprised about half (49.31%) of the consumers visiting the analysed farmers' markets. Wolf and Berrenson [27] found that 41% of consumers had an annual income of less than US $20,000. In their study in the US, Ruelas et al. [37], however, stated that 55% of consumers earned less than US$15,000 per year. In many studies conducted in the US, the net annual household income of the consumers visiting farmers' markets was higher than that of Turkey. The net annual household in-come found by Govindasamy et al. [6] was US$60,000, by Henneberry et al. [4] was US$40,000–60,000, by Conner et al. [12] was US$56,000 and by Wilson et al. [13] was over US$50.000.

When the household size of consumers shopping at farmers' markets was analysed, a small household size of approximately three people (2.92 people) was encountered. In previous studies, it was determined that the average household size of consumers shopping at farmers' markets varied between two or three people [6, 12, 36].

When the employment status of consumers shopping at farmers' markets was examined, three different groups emerged—employees, retirees and housewives. The proportion of employees in these groups was 40.77%, the ratio of retirees was 34.99% and the ratio of housewives was 20.94%.

## Consumer segments of farmers' markets

To reveal the consumer profile of the farmers' markets examined, consumer segments were determined. In the first stage, reliability analysis was employed to assess the reliability of the respondents' answers. In this context, the answers given to the 22 Likert-type scale questions revealed the level of importance that consumers placed on various market attributes. In the analysis, the Cronbach's alpha value was found to be 0.764. As this value is between $0.60 \leq \alpha < 0.80$, the scale is highly reliable.

Before conducting the factor analysis, the effect levels of various market attributes in the decision of consumers to shop at farmers' markets were examined (Table 2). The most important attribute for consumers in the decision to shop at farmers' markets was deter-mined to be product freshness. Product freshness was also found to be one of the main market attributes that motivated consumers to shop at farmers' markets in many previous studies [6–8, 10, 15–23].

Product quality was determined as the second most important attribute that influenced consumers' decision to shop at farmers' markets. Many consumers who shop at farmers' markets believe that the products sold are of good quality. Previous studies have also confirmed that quality is an important source of motivation for consumers visiting farmers' markets [4–13, 14].

Product flavour or taste of farmers' market products has been identified as the third most important attribute that influences consumers' decision to shop at farmers' markets. In Conner et al.'s [17] study in Michigan, US, taste was shown to be among the main motivating factors.

The cultivation using less chemicals and food safety suitability of the products offered in the market are other important attributes that affect consumers' decision to shop at farmers' markets. In fact, it is more appropriate to address these two attributes together as food safety concerns. In many previous studies, food safety concerns have attracted attention as an important market attribute that influences consumers to patronise farmers' markets. Trobe [24]

**Table 2. The importance of various market attributes in consumer decisions to patronise farmers' markets.**

| Market Attribute | $\bar{x}$ | SD |
|---|---|---|
| Freshness of the products | 4.87 | .408 |
| Quality of the products | 4.78 | .573 |
| Taste of the products | 4.77 | .593 |
| Wide variety of products | 3.73 | 1.173 |
| Product availability | 3.44 | 1.335 |
| Availability of locally grown produce | 4.05 | 1.242 |
| Availability of GlobalGAP-certified products | 4.36 | 1.069 |
| Availability of certified organic products | 4.48 | .981 |
| Food safety practices | 4.63 | .717 |
| Foods containing few or no chemicals | 4.72 | .716 |
| Product presentation to buyers | 3.57 | 1.386 |
| Outward appearance of the farmers' market | 3.62 | 1.350 |
| Overall cleanliness of the farmers' market | 4.49 | .887 |
| Availability of entertainment and social activities | 2.10 | 1.460 |
| Transportation facilities to the farmers' market | 3.26 | 1.588 |
| Distance to farmers' markets | 3.16 | 1.594 |
| Availability of parking near and around the farmers' market | 3.61 | 1.646 |
| Availability of food labelling information | 3.75 | 1.307 |
| Operating days and hours of the farmers' market | 2.66 | 1.656 |
| Consumer interactions and vendor behaviours | 4.50 | .832 |
| Product prices | 3.58 | 1.345 |
| Sales promotion activities | 2.71 | 1.629 |
| Cronbach's Alpha | 0.764 | |

$\bar{x}$: Mean importance scores given by respondents to farmers' market attribute (1: not important to 5: very important); SD: standard deviation.

emphasised that consumers visit these markets to access organic and non-GMO foods because of health and food safety concerns. Conner et al. [17] stated that consumers visit farmers' markets to access products that do not contain chemicals or pesticides. According to the findings of Conner et al. [12], Feagan and Morris [18] and Carey et al. [19], the expectation of access to healthy and safe food strongly motivates consumers to visit farmers' markets.

When the suitability of the dataset for various market attributes that might affect consumers' decision to shop at farmers' markets for factor analysis was examined, the result of the KMO test was 0.729. A ratio above 0.70 reveals that the dataset shows a very good level of suitability for factor analysis. The fact that the Barlett's test result is also significant indicates that there is a high correlation between the variables, in other words, the dataset is suitable for factor analysis (chi-squared = 2201.073, df = 231, p < 0.001).

As a result of the factor analysis, six factors with eigenvalue statistics greater than one were determined. The first factor explains 11.71% of the total variance; the second factor, 11.49%; the third factor, 10.39%; the fourth factor, 9.28%; the fifth factor, 7.58% and the sixth factor, 7.48%. It was determined that six factors explained 57.93% of the total variance (Table 3).

Names were given to the dimensions obtained by considering the meaning of the items in the factors by making use of the rotated factor loadings. These factors are—Factor 1: Market Atmosphere, Factor 2: Food Safety, Factor 3: Location and Accessibility, Factor 4: Freshness and Quality, Factor 5: Market Events and Promotion and Factor 6: Availability and Variety.

**Table 3. Factor analysis of market attribute variables using varimax rotation.**

| Market Attribute | Factor Loadings | | | | | |
|---|---|---|---|---|---|---|
| | F1 | F2 | F3 | F4 | F5 | F6 |
| Overall cleanliness of the farmers' market | .692 | | | | | |
| Outward appearance of the farmers' market | .684 | | | | | |
| Availability of food labelling information | .622 | | | | | |
| Consumer interactions and vendor behaviours | .565 | | | | | |
| Product presentation to buyers | .458 | | | | | |
| Product prices | .422 | | | | .322 | |
| Availability of parking near and around the farmers' market | .400 | | | | | |
| Availability of GlobalGAP-certified products | | .873 | | | | |
| Availability of certified organic products | | .839 | | | | |
| Foods containing fewer or no chemicals | | .700 | | | | |
| Food safety practices | .398 | .531 | | | | |
| Distance to farmers' markets | | | .902 | | | |
| Transportation facilities to the farmers' market | | | .894 | | | |
| Operating days and hours of the farmers' market | | | .572 | | | |
| Freshness of the products | | | | .807 | | |
| Quality of the products | | | | .799 | | |
| Taste of the products | | | | .672 | | |
| Sales promotion activities | | | | | .745 | |
| Availability of entertainment and social activities | | | | | .741 | |
| Product availability | | | | | | .770 |
| Wide variety of products | | | | | | .719 |
| Availability of locally grown produce | | | | | | .627 |
| Eigenvalue | 3.877 | 2.758 | 2.010 | 1.654 | 1.300 | 1.146 |
| The percentage of total variance explained by each factor | 11.71 | 11.49 | 10.39 | 9.28 | 7.58 | 7.48 |
| The cumulative percentage of the variance explained by all factors | 57.93 | | | | | |

F1: Market atmosphere; F2: food safety; F3: location and accessibility; F4: freshness and quality; F5: market events and promotion; F6: availability and variety

In the second stage of the study, considering the loadings of the six factors obtained from the factor analysis, cluster analysis was performed by applying the k-means clustering method to determine the consumer segments of farmers' markets. In the discriminant analysis conducted to determine the number of clusters in the k-means clustering method, the number of clusters was two according to the Wilk's lambda value, which has the highest significance. In this case, two segments emerged among the 363 consumers inter-viewed. The first segment includes 132 (36.36%) and the second segment includes 231 (63.64%) consumers.

The consumers in the first and second segments were named according to the last cluster centres, which showed the average of the six factors in two clusters. The consumers in the first segment were named 'Conventional Consumers' because they reflected the behaviour of typical consumers on a basic level. Consumers in this segment placed more emphasis on market location and accessibility, product freshness and quality, market events and promotional activities and product availability and variety. The consumers in the second segment were called 'Conscious Consumers' (Table 4). Consumers in the second segment placed emphasis on food safety and attached importance to products with a certificate of good agriculture and organic agriculture or to crops grown using less chemicals. This group, which attached great importance to food safety, also had a high level of awareness because of the importance of some basic attributes that reflected the market atmosphere, such as market hygiene, exterior market

**Table 4. Final cluster centres.**

| Factor | Cluster | |
|---|---|---|
| | **Cluster 1 (Conventional Consumers)** | **Cluster 2 (Conscious Consumers)** |
| Market atmosphere | −.24656 | .14089 |
| Food safety | −.52872 | .30213 |
| Location and accessibility | .26126 | −.14929 |
| Freshness and quality | .27904 | −.15945 |
| Market events and promotion | .77213 | −.44122 |
| Availability and variety | .08116 | −.04638 |

appearance, product information or labelling and how the producers in the market treated customers.

The analysis of variance (ANOVA) results obtained using cluster analysis were used to determine the differences in the factors by clusters. The ANOVA results revealed that there was a significant difference between the clusters at a 1% level of statistical significance for factors other than the availability and variety (Table 5).

In previous studies, the number of consumer segments determined for consumers who shop at farmers' markets varied between three and five [10, 28, 29]. According to this study, there are two consumer segments shopping at farmers' markets. The smaller number of consumer segments in Turkey's farmers' markets can be attributed to the low level of consumer awareness of farmers' markets, which can be explained because farmers' markets are not sufficiently widespread throughout the country. Conversely, in countries with developed farmers' markets, such as the US and the UK, the history of these markets is extensive; therefore, it can be expected that a greater number of consumer segments will exist in these more developed and older farmers' markets.

Although the number of consumer segments in farmers' markets differs by country, there are similarities between consumer segments. For instance, three of the five different consumer segments identified for farmers' markets in the Chicago and Metro East (St. Louis) metropolitan areas of the US by Elepu and Mazzocco [29] were similar to the segments found in this study. When the consumer segments in both studies are compared, Basic Shoppers corresponds to Conventional Consumers and Conscious Consumers to Serious Shoppers and Market Enthusiasts. Additionally, there are consumer segments that are partially similar to the findings of other studies. The Seasonal Shoppers and Utilitarians segments determined by Hunt [10] for Maine farmers' markets in the US are similar to the Conventional Consumer segment in this study. Enthusiasts, one of the segments determined by Arrington et al. [28] for

**Table 5. K-means cluster analysis: ANOVA results.**

| Factor | Cluster | | Error | | F | Sig. |
|---|---|---|---|---|---|---|
| | **Mean Square** | **df** | **Mean Square** | **df** | | |
| Market atmosphere | 12.609 | 1 | .968 | 361 | 13.028 | .000* |
| Food safety | 57.986 | 1 | .842 | 361 | 68.855 | .000* |
| Location and accessibility | 14.159 | 1 | .964 | 361 | 14.694 | .000* |
| Freshness and quality | 16.151 | 1 | .958 | 361 | 16.859 | .000* |
| Market events and promotion | 123.665 | 1 | .660 | 361 | 187.313 | .000* |
| Availability and variety | 1.366 | 1 | .999 | 361 | 1.368 | .243 |

* denotes statistical significance at the 1% level

**Table 6. Demographic characteristics of consumer segments.**

| Characteristic | Category | Conventional Consumers (%) | Conscious Consumers (%) | Chi-squared value | P-value |
|---|---|---|---|---|---|
| **Gender** | Female | 70.45 | 62.77 | 2.197 | 0.168 |
| | Male | 29.55 | 37.23 | | |
| **Age** | 18–24 | 2.27 | 1.73 | 18.946 | 0.002* |
| | 25–34 | 8.33 | 4.33 | | |
| | 35–44 | 25.00 | 10.39 | | |
| | 45–54 | 21.97 | 28.57 | | |
| | 55–64 | 29.55 | 33.77 | | |
| | 65 and over | 12.88 | 21.21 | | |
| | Average age (in years) | 49.88 | 54.46 | | |
| **Marital status** | Married | 82.58 | 80.09 | 0.338 | 0.582 |
| | Single | 17.42 | 19.91 | | |
| **Education** | Illiterate | 2.27 | 0.87 | 13.030 | 0.023** |
| | Primary school | 28.03 | 14.72 | | |
| | Secondary school | 9.09 | 12.99 | | |
| | High school | 21.21 | 27.27 | | |
| | Undergraduate | 34.09 | 34.63 | | |
| | Graduate | 5.30 | 9.52 | | |
| **Monthly household income (TRY)** | 2,000 or less | 28.03 | 23.81 | 3.350 | 0.501 |
| | 2,001–5,000 | 47.73 | 50.22 | | |
| | 5,001–7,500 | 15.91 | 16.02 | | |
| | 7,501–10,000 | 6.06 | 4.33 | | |
| | More than 10,000 | 2.27 | 5.63 | | |
| **Household size (people)** | 1 | 9.85 | 13.85 | 9.660 | 0.047** |
| | 2 | 22.73 | 35.06 | | |
| | 3 | 31.06 | 22.08 | | |
| | 4 | 22.73 | 19.48 | | |
| | 5 and more | 13.64 | 9.52 | | |
| | Average household size (number of people) | 3.15 | 2.79 | | |
| **Employment status** | Retired | 28.03 | 38.96 | 10.480 | 0.033** |
| | Housewife | 28.79 | 16.45 | | |
| | Employed | 40.91 | 40.69 | | |
| | Student | 1.52 | 1.30 | | |
| | Unemployed | .76 | 2.60 | | |

* and ** denote statistical significance at the 1% and 5% levels, respectively.

Indiana and Illinois farmers' markets in the US, are somewhat similar to the Conscious Consumer segment in this study.

In Table 6, the demographic characteristics that characterise the socioeconomic pro-file of consumer segments—gender, age group, marital status, education level, household net income, household size and employment status—are provided for each segment.

The chi-squared test results revealed that there was a statistically significant difference between the two segments in terms of age group, education level, household size and employment status at different probability levels. However, there was no statistically significant difference between the two consumer segments in terms of gender, marital status and average net monthly household income.

The vast majority of conventional consumers and conscious consumers are women and married. Both segments show similar characteristics in terms of net monthly house-hold income. Almost half of the conventional consumers and half of conscious consumers had a monthly income of 2,001–5,000 TRY (US$662.58–1,655.63). As of the period when the data were obtained, the consumers in both segments had a middle income in terms of household income distribution. The ratio of those who have a monthly income of more than 5,000 TRY (US$1,655.63) was 24.24% in the Conventional Consumers segment and 25.98% in the Conscious Consumers segment.

The conventional consumers in the first segment are generally younger and less educated than conscious consumers in the second segment. The household size of the consumers in this segment is higher than in the Conscious Consumers segment. The ratio of employees among conventional consumers is high; the ratio of retirees is at least 10% lower than in the Conscious Consumer segment. Additionally, the ratio of housewives in the Conventional Consumer segment (28.79%) was determined to be approximately 12% higher than in the Conscious Consumer segment (16.45%).

The profile of the Conscious Consumers segment differed from the Conventional Consumers segment. Conscious consumers are generally older and have better education than conventional consumers. The average household size of conscious consumers is smaller than that of conventional consumers. As with conventional consumers, the pro-portion of employees working among conscious consumers is high. However, it is note-worthy that the rate of retirees is quite high compared to the Conventional Consumer segment. Unlike conventional consumers, the proportion of housewives is lower in the Conscious Consumers segment.

## Behavioural characteristics of consumer segments in farmers' markets

Consumer segments were also analysed in terms of various behavioural characteristics, such as the frequency of visiting farmers' markets, number of farmers' markets visit-ed, average expenditure per visit to the farmers' market and distance between the consumer's residence and the farmers' market (Table 7). A statistically significant difference was found between the two consumer segments in terms of the frequency of visiting farmers' markets. It was determined that conscious consumers visit farmers' markets more frequently than conventional consumers. While 73.33% of conscious consumers visited farmers' markets once a week, this rate is 57.81% for conventional consumers. The ratio of consumers visiting farmers' markets at less frequent intervals was higher for the Conventional Consumer segment compared to the Conscious Consumers.

No statistically significant difference was found between consumer segments in terms of the number of farmers' markets visited. While the average number of markets visited by conventional consumers is 1.42, this number is 1.60 for conscious consumers. Although there is no statistically significant difference, conscious consumers in general visit a larger number of farmers' markets. While the ratio of consumers who visit only one farmers' market is 72.66% in conventional consumers, this ratio is 62.67% for conscious consumers. Conversely, there are more farmers' markets visited by consumers in other countries. For instance, in a study done by Govindasamy et al. [6] in the US (New Jersey), it was found that the majority of consumers (67%) visited two to four different farmers' markets. This result can be attributed to the high number of farmers' markets in the US and the high recognition of these markets by consumers as they have a longer history. Depending on the development of farmers' markets, the number of markets visited is anticipated to also increase in Turkey.

No statistically significant difference was found between the consumer segments in terms of the average amount spent at farmers' markets per visit. While conventional consumers

**Table 7. Behavioural characteristics of consumer segments.**

| Characteristic | Category | Conventional Consumers (%) | Conscious Consumers (%) | Chi-squared value | P-value |
|---|---|---|---|---|---|
| The frequency of consumer visits to farmers' markets | Once a week | 57.81 | 73.33 | 21.291 | 0.006* |
| | Two times a week | 10.94 | 8.89 | | |
| | Three times a week | - | 0.44 | | |
| | Every other week | 11.72 | 6.22 | | |
| | Once a month | 10.16 | 8.89 | | |
| | Once a year | 4.69 | - | | |
| | Twice a year | 3.91 | 1.78 | | |
| | Three times in one year | 0.78 | - | | |
| | Seven times a year | - | 0.44 | | |
| Number of farmers' markets visited | 1 | 72.66 | 62.67 | 7.665 | 0.176 |
| | 2 | 18.75 | 22.22 | | |
| | 3 | 3.91 | 10.67 | | |
| | 4 | 3.13 | 2.22 | | |
| | 5 | 1.56 | 1.33 | | |
| | 6 | - | 0.89 | | |
| | Average number of farmers' markets visited | 1.42 | 1.60 | | |
| The amount spent at farmers' markets per visit (in TRY) | ≤25 | 26.56 | 16.00 | 5.883 | 0.117 |
| | 26–50 | 41.41 | 45.33 | | |
| | 51–75 | 12.50 | 15.56 | | |
| | >75 | 19.53 | 23.11 | | |
| | Average amount spent per visit (TRY) | 52.32 | 59.56 | | |
| Distance between consumers' residences and the farmers' markets (in kilometres) | <1 | 50.00 | 29.00 | 18.934 | 0.002* |
| | 1–5 | 21.21 | 38.53 | | |
| | 6–10 | 8.33 | 10.39 | | |
| | 11–20 | 4.55 | 3.90 | | |
| | 21–30 | 2.27 | 1.73 | | |
| | >30 | 13.64 | 16.45 | | |

* and ** denote statistical significance at the 1% and 5% levels, respectively.

spent an average of 52.32 TRY (US$17.32) each time they visited a farmers' market, conscious consumers spent 59.56 TRY (US$19.72). In both consumer segments, the percentage of those who spent between 26 and 50 TRY (US$8.61–16.56) per market visit is above 40%. Even though there is not much difference between them, conscious consumers spent a little more per market visit. While the ratio of those who spent 25 TRY (US$8.28) or less per market visit is 26.56% in conventional consumers, this rate is 16% in conscious consumers. In the US, the average amount spent by consumers per visit to farmers' markets was US$20.44 in Illinois [29], US$15–30 in Los Angeles [37] and US$24.78 in Nevada and Utah [20]. In a study conducted in Italy, however, the average spending of consumers was determined as €22.51 [36]. According to these results, the average amount spent by conventional and conscious consumers per visit in farmers' markets is consistent with the findings of previous studies.

Considering how far the consumers travel from where they live to go to farmers' markets, there is a statistically significant difference between the two consumer segments. In general, the proportion of consumers coming from close distance was high in conventional consumers. The proportion of people who travelled less than one kilometre (km) to patronise a farmers'

market is 50% in the Conventional Consumer segment, whereas this rate is 29% in the Conscious Consumer segment. The ratio of consumers coming to farmers' markets from a distance of 15 km is higher in conscious consumers (38.53%) than in conventional consumers (21.21%). It is noteworthy that conscious consumers come a long way to patronise farmers' markets. This is because conscious consumers value farmers' markets more, especially in terms of food safety.

## Conclusions

The findings of this study make an important contribution to the literature in terms of revealing the consumer profile in countries wherein farmers' markets are in the development stage. This study confirms the hypothesis that the factors that motivate consumers to shop at farmers' markets vary according to consumer segment. Two consumer segments—Conventional Consumers and Conscious Consumers—were determined in the examined farmers' markets. Conventional consumers reflect the behaviour of typical consumers on a basic level, whereas conscious consumers represent a group that is more sensitive about food safety.

According to the findings of this study, the majority of consumers (63.64%) who visited farmers' markets were from the Conscious Consumer segment. This makes it necessary to manage farmers' markets more effectively. Conscious consumers represent the group most loyal to farmers' markets. The most important motivating factor that brings conscious consumers to farmers' markets is food safety concerns. Even if they know that they cannot find certificated products produced with organic or good agricultural methods at farmers' markets, consumers assume that the products offered for sale are grown using less chemicals. Conscious consumers believe that the vendors at farmers' markets are controlled by the relevant local municipalities; therefore, healthier products are offered in these markets. However, how healthy the products sold by farmers' markets vendors actually are in terms of food safety is controversial. Although farmers' markets operate under the control of local municipalities, there are many deficiencies. Most importantly, the vendors selected to sell at farmers' markets are not chosen carefully. Local municipalities should evaluate vendors in terms of different criteria, especially the suitability for healthy production conditions, when choosing the vendors for farmers' markets. Doubts regarding how healthy the products offered in farmers' markets are can never be eliminated without implementing a certification system. In this context, it should be obligatory to obtain certification for products offered for sale directly by vendors at farmers' markets. The findings obtained from the interviewed consumers revealed that there are other important problems to be resolved in farmers' markets. First, the physical structure of farmers' markets should be improved and these markets should be made more visual. Additionally, it is crucial to make the farmers' markets an entertaining environment featuring social activities such as events, concerts and festivals.

In Turkey, there is no specific legal regulation that covers only farmers' markets. The current legal regulations have been prepared for both farmers' and street markets. However, farmers' markets are different and more specialised than street markets. Therefore, legal regulations regarding farmers' markets should be handled using a different framework.

The results of this study can serve as a reference that presents scientific findings to help guide existing and prospective farmers' markets. Thus, those who manage farmers' markets can ensure that consumer demands are more effectively met by these markets. This might lead to an increase in the number of consumers patronising existing farmers' markets and in the demand for new farmers' markets.

Farmers' markets in Turkey are generally located in districts around the province. This situation limits the scope of studies on the subject. As a matter of fact, seven of the eight farmers'

markets examined in this study are located in districts. However, attempts by municipalities to establish farmers' markets in urban centres have been increasing of late. In this framework, it is suggested that future studies should be conducted within the scope of farmers' markets in urban centres.

This research was carried out about five years ago. There may have been a change in consumer behavior at farmers' markets since then. Also, the COVID-19 pandemic may have changed customers' shopping behaviour. Therefore, it is recommended to consider both situations in future research.

## Supporting information

**S1 Data.**
(XLSX)

## Acknowledgments

The author is grateful to Ege University Planning and Monitoring Coordination of Organizational Development and Directorate of Library and Documentation for their support regarding the editing and proofreading service for this study. The author also wishes to thank the consumers who contributed to the farmers' market survey in the Izmir, Turkey.

## Author Contributions

**Conceptualization:** Hakan Adanacioglu.

**Formal analysis:** Hakan Adanacioglu.

**Funding acquisition:** Hakan Adanacioglu.

**Investigation:** Hakan Adanacioglu.

**Methodology:** Hakan Adanacioglu.

**Project administration:** Hakan Adanacioglu.

**Visualization:** Hakan Adanacioglu.

**Writing – original draft:** Hakan Adanacioglu.

**Writing – review & editing:** Hakan Adanacioglu.

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
