## [Decision Letter · Decision Letter 0]

19 May 2021

PONE-D-21-10649

Factors affecting the purchase behaviour of farmers’ markets consumers: A case study in Izmir, Turkey

PLOS ONE

Dear Dr. Adanacioglu,

Thank you for submitting your manuscript to PLOS ONE. After careful consideration, we feel that it has merit but does not fully meet PLOS ONE’s publication criteria as it currently stands. Therefore, we invite you to submit a revised version of the manuscript that addresses the points raised during the review process.

We look forward to receiving your revised manuscript.

Kind regards,

Prof. Arkadiusz Piwowar

Wroclaw University of Economics and Business

Academic Editor

PLOS ONE

Reviewers' comments:

Reviewer #1: Dear Author, the research provides some useful information regarding factors which motivate consumers to buy food products from farmers' markets. The article is interesting but some more in-depth analysis has to be done and its structure has to be modified.

1. The article’s aim is to: "determine the factors that motivate consumers who shop at farmers’ markets". Nevertheless, I’m still not convinced why such study is needed. You should state clearly why it is an interesting topic to research into.

2. The use of "a case study..." phrase in the title is misleading, as it suggests the use of the case study methodology which was not applied to the research. The article lacks well-formulated hypotheses and their theoretical justification. The research was carried out quite a long time ago (5 years), so it can be presumed that consumers’ behaviors may have changed since then. That fact should be emphasized in the paper.

3. The study of consumers' motivation making them buy goods at farmers’ market was limited by the Author only to a small number of factors which were poorly defined, like, for example, in the case of the quality of food. It is a pity that the issues related to consumers’ values, which are connected to food choice, health, socialization, the asymmetry of information or consumer knowledge, have been omitted. What is more, there was no division between urban and rural markets. Is it because only urban markets were analyzed? Were there any tourists among respondents choosing a particular market’s offer due to culinary tourism?

4. The article is presented in an easy to understand way. The prepared statistical analysis of the results is simple. It is worth considering shortening the text, especially in relation to the parts that contain the description of the tables included (this is an unnecessary duplication of information).

5. The characteristics of the studied sample should not be included in the "Results and discussion" section. I think the Author needs to develop a separate section concerning the “discussion”. The paper lacks such parts as: conclusion, limitations and future research.

6. Another thing worth considering are the names of factors (factor analysis), because they seem to be inadequate for the items. Perhaps they should be called differently.

7. The extracted segments were characterized by socio-demographic variables. Did the collected test results not give the possibility to carry out a more in-depth analysis?

8. It is also worth enriching the preliminary considerations on farmers’ markets by comparing them in countries at a similar stage of development of such form of food distribution.

9. Overall, the manuscript is quite readable. Nevertheless, there are still some linguistic challenges regarding punctuation and, to some extent, grammar.

Reviewer #2: The paper is in line with the aims and scope of the journal. The research was clearly explained.

Furthermore, I suggest a little improvement of bibliography in the first section of the paper, and how did the author selected the farmers market. The author did not explain how did he develop the instrument, did you do pilot? Why did you deleted 21 cases, they might provided important information.

Comment: In the same way, conclusions appear poor especially because of the repeat results

discussed in the previous sections. I suggest improving them with future prospects of research and by

discussing the implications that these strategies could have in the future (in terms of crop choices in market,

environmental, social and economic implications and so on. Also would useful if you include the government policy regarding good agriculture practices.

Comment: I do not understand why author did not use regression analysis in the study.

---

## [Author Response · Author response to Decision Letter 0]

12 Jun 2021

Response to Reviewer 1 Comments

Point 1: The article’s aim is to: "determine the factors that motivate consumers who shop at farmers’ markets". Nevertheless, I’m still not convinced why such study is needed. You should state clearly why it is an interesting topic to research into. 

Response 1: First of all, thank you for your valuable comments. There are some important reasons that increase the importance of this study. Previous studies on farmers’ markets mostly focus on North America. Additional research is required that examines farmers’ markets in different parts of the world. It is especially important to examine countries in which farmer markets are just beginning to develop. Turkey is one of the countries in which farmers’ markets have begun to become widespread in recent years.

In most of previous studies, the factors affecting the shopping motivation of consumers in farmer markets were examined. However, it is observed that there are very few studies on determining the consumer segments in farmer markets. This study also reveals consumer segments in the farmers market in Turkey. Farmers market in Turkey has started new development. More studies are needed on this subject. Therefore, I think this study will make an important contribution to the literature.

Point 2: The use of "a case study..." phrase in the title is misleading, as it suggests the use of the case study methodology which was not applied to the research. The article lacks well-formulated hypotheses and their theoretical justification. The research was carried out quite a long time ago (5 years), so it can be presumed that consumers’ behaviors may have changed since then. That fact should be emphasized in the paper.

Response 2:

- The phrase “A case study..” in the title was written to emphasize that this study was carried out in the Izmir region of Turkey. Based on your suggestion, this phrase has been removed from the title.

- The hypothesis of this article was stated in the introduction (lines 104-109):

Previous studies indicate that there are various types of customer segments shopping at farmers’ markets [10,26,27]. Considering previous studies, the following research hypothesis was proposed:

H1. The factors that motivate consumers to shop at farmers’ markets vary according to various consumer segments.

- The judgment of the proposed hypothesis was made between lines 376 and 396.

- You emphasized that consumer behavior may change because this research was conducted five years ago. Considering your opinion, the following paragraph has been added to the conclusion part (lines 556-559).

“This research was carried out about five years ago. There may have been a change in consumer behavior at farmers' markets since then. Also, the COVID-19 pandemic may have changed customers' shopping behaviour. Therefore, it is recommended to consider both situations in future research.”

Point 3: The study of consumers' motivation making them buy goods at farmers’ market was limited by the Author only to a small number of factors which were poorly defined, like, for example, in the case of the quality of food. It is a pity that the issues related to consumers’ values, which are connected to food choice, health, socialization, the asymmetry of information or consumer knowledge, have been omitted. What is more, there was no division between urban and rural markets. Is it because only urban markets were analyzed? Were there any tourists among respondents choosing a particular market’s offer due to culinary tourism? 

Response 3:

-Thank you very much for your valuable comments. While determining the 22 variables used in the analysis, the main motivation factors examined for the consumers of farmer markets in previous studies were taken into consideration (see Table 2). A significant portion of the consumer values connected with food preference was included in the analysis. One of the main motivation factors obtained as a result of factor analysis is the "food safety" factor under the code F2. This factor includes consumer concerns about health. The "market events and promotion" factor under the F5 code includes socialization.

-No distinction was made between rural and urban farmers' markets for comparison. Because seven of the eight farmers' markets where this research was conducted are located in rural areas. An explanation is given in the conclusion section regarding this issue (lines 551-556).

“Farmers’ markets in Turkey are generally located in districts around the province. This situation limits the scope of studies on the subject. As a matter of fact, seven of the eight farmers’ markets examined in this study are located in districts. However, attempts by municipalities to establish farmers’ markets in urban centres have been increasing of late. In this framework, it is suggested that future studies should be conducted within the scope of farmers’ markets in urban centres.”

-Among the participants, there was no one who came especially because of culinary tourism.

Point 4: The article is presented in an easy to understand way. The prepared statistical analysis of the results is simple. It is worth considering shortening the text, especially in relation to the parts that contain the description of the tables included (this is an unnecessary duplication of information). 

Response 4: You are right that the analytical procedures are a bit too much in the article. This extends the text of the article. This article is the first comprehensive academic study of the farmers' markets in Turkey. For this reason, I think it may be useful for readers to display these descriptive statistics.

Point 5: The characteristics of the studied sample should not be included in the "Results and discussion" section. I think the Author needs to develop a separate section concerning the “discussion”. The paper lacks such parts as: conclusion, limitations and future research.

Response 5: 

- According to the journal rules, the “Results and discussion” section can be combined together. The “Results and discussion” section includes both the results of this survey and the discussion on these results.

-The "Conclusions" section of this study includes parts of "conclusion", "limitations" and "future research".

Point 6: Another thing worth considering are the names of factors (factor analysis), because they seem to be inadequate for the items. Perhaps they should be called differently.

Response 6: In factor analysis, I took into account the market attributes while giving the factor names. Also, I reviewed previous studies. I think these factor names are in line with the market attributes.

Point 7: The extracted segments were characterized by socio-demographic variables. Did the collected test results not give the possibility to carry out a more in-depth analysis? 

Response 7: In this study, factor analysis was performed before determining the consumer segments. Cluster analysis was performed using the data obtained from factor analysis. After the cluster analysis, consumer segments in farmers' markets were determined. Then, the descriptive features of the identified consumer segments were examined. Similar methods have been used in previous studies [10,26,27]. In general, after factor analysis, either clustering or regression analysis is performed. Cluster analysis was performed in this study. In future studies, in depth analysis can be made on the subject.

Point 8: It is also worth enriching the preliminary considerations on farmers’ markets by comparing them in countries at a similar stage of development of such form of food distribution.

Response 8: Previous studies of farmers' markets have generally focused on North America. However, in this study, all previous studies on the subject were examined. The findings of these studies are stated both in the “introduction” and in the “results and discussion” section. I have prepared a table summarizing the studies carried out in various countries on the subject. This table is very detailed and large. Therefore, I did not give this table in the text. Instead, I have shown the findings in the table in the text. Please see the attached file for the table.

Point 9: Overall, the manuscript is quite readable. Nevertheless, there are still some linguistic challenges regarding punctuation and, to some extent, grammar. 

Response 9: This manuscript has been edited for English language and spelling by Enago, an editing brand of Crimson Interactive Inc. The certificate sent by Enago has been uploaded to the journal's online submission system. However, if you see a linguistic problem for the article, I can have it checked again. 

Response to Reviewer 2 Comments

Point 1: The paper is in line with the aims and scope of the journal. The research was clearly explained. Furthermore, I suggest a little improvement of bibliography in the first section of the paper.

Response 1: First of all, thank you for your valuable comments. The bibliography has been improved based on your suggestion. Two more references have been added to previous studies on the subject. I have prepared a table summarizing the studies carried out in various countries on the subject. This table is very detailed and large. Therefore, I did not give this table in the text. Instead, I have shown the findings in the table in the text. Please see the attached file for the table.

Point 2: how did the author selected the farmers market. The author did not explain how did he develop the instrument, did you do pilot? Why did you deleted 21 cases, they might provided important information. 

Response 2: 

-The method followed in the selection of farmers' markets where this research was conducted was explained in the first paragraph of “Materials and methods”. This explanation is shown below.

The line numbers in the revised manuscript:129-134.

“In selecting the farmers’ markets where the research was conducted, the active working status of the markets and the fact that the vendors were producers were considered. In this context, eight farmers’ markets in seven districts, including Seferihisar, Urla, Çeşme, Foça, Aliağa, Bergama and Bornova, were included in the research area. Two of the eight farmers’ markets included in the research are located in the Seferihisar district (i.e. Merkez and Sığacık).”

-I had to delete 21 cases. Because these cases contained insufficient information. At the same time, the data in these cases were inconsistent in terms of data reliability. Explanation on this subject is given in the material and method section (lines 161-163).

“However, 21 of these surveys wherein the researchers were given incomplete information and data reliability was suspected were not included in the data analysis. In this context, a total of 363 surveys were considered for data analysis of eight farmers’ markets.”

Point 3: In the same way, conclusions appear poor especially because of the repeat results discussed in the previous sections. I suggest improving them with future prospects of research and by discussing the implications that these strategies could have in the future (in terms of crop choices in market, environmental, social and economic implications and so on. Also would useful if you include the government policy regarding good agriculture practices. 

Response 3: 

-Future strategies for farmers' markets are outlined in the conclusion (lines 522-547). Attention was drawn to the importance of certification in product selection in farmers' markets. Besides; Suggestions were presented on the selection of vendors in farmers' markets, the physical structure of the market and the management of farmers' markets.

-The purpose of this study is not to discuss good agricultural practices. It is considered important that the products offered for sale in farmers' markets are produced with healthy production methods such as organic agriculture or good agriculture. For this purpose, various suggestions have been presented.

“According to the findings of this study, the majority of consumers (63.64%) who vis-ited farmers’ markets were from the Conscious Consumer segment. This makes it necessary to manage farmers’ markets more effectively. Conscious consumers represent the group most loyal to farmers’ markets. The most important motivating factor that brings conscious consumers to farmers’ markets is food safety concerns. Even if they know that they cannot find certificated products produced with organic or good agricultural methods at farmers’ markets, consumers assume that the products offered for sale are grown using less chemicals. Conscious consumers believe that the vendors at farmers’ markets are controlled by the relevant local municipalities; therefore, healthier products are offered in these markets. However, how healthy the products sold by farmers’ markets vendors actually are in terms of food safety is controversial. Although farmers’ markets operate under the control of local municipalities, there are many deficiencies. Most importantly, the vendors selected to sell at farmers’ markets are not chosen carefully. Local municipalities should evaluate vendors in terms of different criteria, especially the suitability for healthy production conditions, when choosing the vendors for farmers’ markets. Doubts regarding how healthy the products offered in farmers’ markets are can never be eliminated without implementing a certification system. In this context, it should be obligatory to obtain certification for products offered for sale directly by vendors at farmers’ markets. The findings obtained from the interviewed consumers revealed that there are other important problems to be resolved in farmers’ markets. First, the physical structure of farmers’ markets should be improved and these markets should be made more visual. Additionally, it is crucial to make the farmers’ markets an entertaining environment featuring social activities such as events, concerts and festivals. In Turkey, there is no specific legal regulation that covers only farmers’ markets. The current legal regulations have been prepared for both farmers’ and street markets. However, farmers’ markets are different and more specialised than street markets. Therefore, legal regulations regarding farmers’ markets should be handled using a different framework.”

- Examining the environmental, social and economic impacts of farmers' markets is another research topic. Warsaw et al. (2021)'s study titled “The Economic, Social, and Environmental Impacts of Farmers Markets: Recent Evidence from the US” is a specific example made in this context. However, considering your suggestion, a study for this purpose can be carried out within the scope of Turkey in future studies.

Warsaw, P.; Archambault, S.; He, A.; Miller, S. The Economic, Social, and Environmental Impacts of Farmers Markets: Recent Evidence from the US. Sustainability 2021, 13, 3423. https://doi.org/10.3390/su13063423

Point 4: I do not understand why author did not use regression analysis in the study

Response 4: In this study, factor analysis was performed before determining the consumer segments. Cluster analysis was performed using the data obtained from factor analysis. After the cluster analysis, consumer segments in farmers' markets were determined. Then, the descriptive features of the identified consumer segments were examined. Similar methods have been used in previous studies [10,26,27]. In general, after factor analysis, either clustering or regression analysis is performed. Cluster analysis was performed in this study. In future studies, in depth analysis can be made on the subject.

---

## [Decision Letter · Decision Letter 1]

16 Jul 2021

Factors affecting the purchase behaviour of farmers’ markets consumers

PONE-D-21-10649R1

Dear Dr. Adanacioglu,

We’re pleased to inform you that your manuscript has been judged scientifically suitable for publication and will be formally accepted for publication once it meets all outstanding technical requirements.

Kind regards,

Arkadiusz Piwowar

Wroclaw University of Economics and Business

Academic Editor

PLOS ONE

Reviewers' comments:

Reviewer's Responses to Questions

**Comments to the Author**

1. If the authors have adequately addressed your comments raised in a previous round of review and you feel that this manuscript is now acceptable for publication, you may indicate that here to bypass the “Comments to the Author” section, enter your conflict of interest statement in the “Confidential to Editor” section, and submit your "Accept" recommendation.

Reviewer #1: All comments have been addressed

Reviewer #2: All comments have been addressed

6. Review Comments to the Author

Reviewer #2: The author has significantly improved the paper taking into account all the comments and suggestions. The additions are very useful, and provide additional information that may be critical for the practices in the region. I have no objection to the additions, and the paper may be accepted for publication in its current form.

---

## [Editor Report · Acceptance letter]

21 Jul 2021

PONE-D-21-10649R1 

Factors affecting the purchase behaviour of farmers’ markets consumers 

Dear Dr. Adanacioglu:

I'm pleased to inform you that your manuscript has been deemed suitable for publication in PLOS ONE. Congratulations! Your manuscript is now with our production department. 

Kind regards, 

on behalf of

Professor Arkadiusz Piwowar 

Academic Editor

PLOS ONE